# GraphRCG: Self-Conditioned Graph Generation

## Abstract

Graph generation aims to create new graphs that closely align with a target graph distribution. Existing works often implicitly capture this distribution by aligning the output of a generator with each training sample. As such, the overview of the entire distribution is not explicitly captured and used for graph generation. In contrast, in this work, we propose a novel self-conditioned graph generation framework designed to explicitly model graph distributions and employ these distributions to guide the generation process. We first perform self-conditioned modeling to capture the graph distributions by transforming each graph sample into a low-dimensional representation and optimizing a representation generator to create new representations reflective of the learned distribution. Subsequently, we leverage these bootstrapped representations as self-conditioned guidance for the generation process, thereby facilitating the generation of graphs that more accurately reflect the learned distributions. We conduct extensive experiments on generic and molecular graph datasets. Our framework, GraphRCG, demonstrates superior performance over existing state-of-the-art graph generation methods in terms of graph quality and fidelity to training data.

## 1 Introduction

The task of generating graphs that align with a specific distribution plays a crucial role in various fields such as drug discovery (Shi et al., 2019), public health (Guo et al., 2021), and traffic modeling (Yu & Gu, 2019). In recent times, deep generative models have been prevalently studied to address the problem of graph generation Lee et al. (2023); Jo et al. (2022); Luo et al. (2021). Unlike conventional methods that rely on random graph models, recent methods generally learn graph distributions through advanced deep generative models, e.g., variational autoencoders (VAEs) (Guo et al., 2020; Wang et al., 2022), generative adversarial networks (GANs) (Gamage et al., 2020; De Cao & Kipf, 2018), normalizing flows (Zang & Wang, 2020; Luo et al., 2021), and diffusion models (Lee et al., 2023; Niu et al., 2020; Vignac et al., 2022). These models excel at capturing complex structural patterns in graphs, enabling the creation of new graphs with desirable characteristics.

Despite these advances, the precise modeling and utilization of graph distributions, although crucial for high-fidelity generation, remains underexplored. In fact, it is essential to accurately capture and utilize important patterns in the training data for generation (Kong et al., 2023; Karami, 2023), particularly in complex scenarios like molecular graph generation (Du et al., 2022). For example, precise modeling of molecular properties is key to optimizing molecular structures while maintaining similarity to known molecules. However, the prevalent strategy is to use reconstruction loss to implicitly embed graph distribution within the generator, which may compromise effectiveness. Moreover, the utilization of the captured graph distributions is also less investigated. Ideally, generators should be designed to explicitly guide the generative process, ensuring that the output graphs closely follow the defined graph distributions. Nonetheless, existing research tends to rely on simple features to control generation, such as molecular characteristics (Vignac et al., 2022) or degree information (Chen et al., 2023). Such a strategy requires domain knowledge to design the specific features, while also lacking more comprehensive modeling of the entire distribution. Therefore, the study of graph generation is confronted with two crucial research questions (*RQ*s): (**RQ1**) **Capturing Distributions.** How to precisely capture the graph distribution with rich information helpful for the generation process? (**RQ2**) **Utilizing Distributions.** How to adeptly harness these distributions as direct guidance for the generation of graphs? Addressing these challenges is essential for high-fidelity graph generation.

In practice, however, the above research questions present significant challenges due to the intricate nature of graph data. (1) **Complex Dataset Patterns.** Real-world graphs, such as social networks (Chen et al., 2023) and molecular structures (Lee et al., 2023), exhibit highly complex patterns. These include varying degrees of sparsity, inconsistent clustering coefficients, and specific distributions of node and edge attributes (Huang et al., 2022). Capturing these complex patterns accurately through generative models can be particularly challenging. (2) **Progressive Alignment to Training data**. Unlike images, where generation is often treated as a pixel-wise or patch-wise process (Ho et al., 2022; Rombach et al., 2022; Dhariwal & Nichol, 2021), the generation of graphs is inherently sequential (You et al., 2018b; Wang et al., 2022; Niu et al., 2020). That being said, graphs are generated through a sequence of steps, each with significant implications, such as modifying chemical properties via the addition or removal of atoms and bonds (Shi et al., 2019; Kong et al., 2023; Karami, 2023). As a result, it is subop-

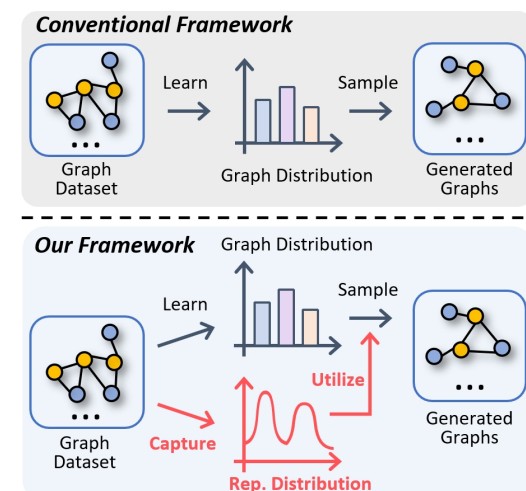

Figure 1: The comparison between the conventional framework and ours. Instead of directly learning the graph distribution, we encode graphs into representations, and learn their distributions for further utilization during generation.

timal to directly guide generation toward true distributions, particularly in the initial steps of the process with graphs largely deviating from the learned distribution.

To deal with these two challenges, in this work, we introduce a novel graph generation framework named GraphRCG, which targets at **Graph R**epresentation-**C**onditioned **G**eneration. As presented in Fig. 1, our framework is designed to first encode graphs into representations, and then capture and utilize such representation distributions for graph generation. By operating on representations instead of directly on graphs, we manage to effectively distill complex, dataset-specific knowledge into these representations and also enable their further utilization for graph generation. GraphRCG is built upon two integral components: (1) **Self-Conditioned Modeling.** To capture graph distributions while addressing the issue of complex dataset patterns in RQ1, we propose to model the essence of graph distribution through a representation generator, which could produce bootstrapped representations that authentically reflect the learned distribution. This strategy is able to enhance the quality of captured graph distributions by encapsulating the complex patterns in a parametrized manner. Moreover, the design also enables the subsequent utilization of captured distributions for generation through sampling diverse representations. (2) **Self-Conditioned Guidance.** We utilize the acquired distributions to guide graph generation to ensure the fidelity of generated graphs regarding the learned distributions. Regarding RQ2, to overcome the challenge of discrete sequential generation, we introduce a novel strategy of step-wise guidance. This strategy employs bootstrapped representations with varying degrees of noise throughout different steps of graph generation, guiding each step closer to the learned distributions in a progressive manner while obviating the need for additional human intervention. In summary, our contributions are as follows:

- In this work, we explore the potential and importance of explicitly capturing and utilizing training data distributions for graph generation to enhance performance by generating graphs that are more closely aligned with the training distributions.

- We innovatively propose a self-conditioned graph generation framework to capture and utilize training data distributions via bootstrapped representations with our devised self-conditioned modeling and self-conditioned guidance, respectively.

- We perform a systematic study to evaluate the performance of our framework in a variety of real-world and synthetic datasets. The results demonstrate the effectiveness of our framework in comparison to other state-of-the-art baselines.

## 2 RELATED WORKS

### 2.1 DENOISING DIFFUSION MODELS FOR GENERATION

Recently, denoising diffusion models (Sohl-Dickstein et al., 2015; Ho et al., 2020; Song et al., 2021) have been widely adopted in various fields of generation tasks. Their notable capabilities span various domains, including image generation (Dhariwal & Nichol, 2021; Rombach et al., 2022; Ho et al., 2022), text generation (Austin et al., 2021; Li et al., 2022; Chen et al., 2022), and temporal data modeling (Tashiro et al., 2021; Lopez Alcaraz & Strodthoff, 2023). In general, denoising diffusion models are probabilistic models designed to learn a data distribution by denoising variables with normal distributions (Song et al., 2020; Kong & Ping, 2021). In particular, they first create noisy data by progressively adding and intensifying the noise in the clean data, in a Markovian manner. Subsequently, these models learn a denoising network to backtrack each step of the perturbation. During training, the denoising networks are required to predict the clean data or the added noise, given the noisy data after perturbation. After optimization, the denoising networks could be used to generate new data via iterative denoising of noise sampled from a prior distribution (San-Roman et al., 2021; Vahdat et al., 2021).

### 2.2 GRAPH GENERATION

Based on the strategies used, graph generation methods could be classified into two categories: (1) *One-shot Generation*. In this category, the models generate all edges among a defined node set in one single step. One-shot generation models are typically built upon the Variational Autoencoder (VAE) or the Generative Adversarial Network (GAN) structure, aiming to generate edges independently based on the learned latent embeddings. Normalizing flow models (Zang & Wang, 2020; Luo et al., 2021) propose to estimate the graph density, by establishing an invertible and deterministic function to map latent embeddings to the graphs. More recently, diffusion models have also been adopted for graph generation (Lee et al., 2023; Jo et al., 2022). To deal with the discrete nature of graph data, DiGress (Vignac et al., 2022) leverages discrete diffusion by considering node and edge types as states in the Markovian transition matrix. (2) *Sequential Generation*. This strategy entails generating graphs through a series of sequential steps, typically by incrementally adding nodes and connecting them with edges. Models in this category often utilize recurrent networks (Li et al., 2018; You et al., 2018b) or Reinforcement Learning (RL) (You et al., 2018a) to guide the generation process (Shi et al., 2019; Ahn et al., 2021). Sequential generation is particularly suitable for generating graphs with specific desired properties (Zhu et al., 2022).

### 2.3 CONDITIONAL GENERATION

Recent works have also explored various strategies to condition generation on specific classes or features. For example, ARM (Dhariwal & Nichol, 2021) proposes to utilize gradients from classifiers to guide the generation process within each step. LDM (Rombach et al., 2022) enables the incorporation of external information, such as text (Reed et al., 2016) and semantic maps (Isola et al., 2017), with a specific encoder to learn the representations. The representations are then incorporated into the underlying UNet backbone (Ronneberger et al., 2015). In RCG (Li et al., 2023), the authors employ a self-conditioned strategy to condition generation on representations learned from a pre-trained encoder. Despite the advancements in image generation, it still presents significant difficulty when applying these methods in graph generation, due to the complex dataset patterns (Chen et al., 2023; Lee et al., 2023; Huang et al., 2022) and discrete sequential generation (Wang et al., 2022; Niu et al., 2020; Shi et al., 2019). In contrast, our framework deals with these challenges with self-conditioned modeling and guidance to enhance graph generation efficacy.

## 3 GRAPHRCG: SELF-CONDITIONED GRAPH GENERATION

Our self-conditioned graph generation framework consists of two modules: self-conditioned modeling and self-conditioned guidance for RQ1 and RQ2, respectively. As illustrated in Fig. 2, in self-conditioned modeling, we employ a representation generator to capture graph distributions by learning to denoise representations with noise. The alignment loss between noisy representations and noisy graphs acts as a self-supervised loss to train the encoder. With the optimized representation generator, we train the graph generator by denoising graphs with added noise. At each generation

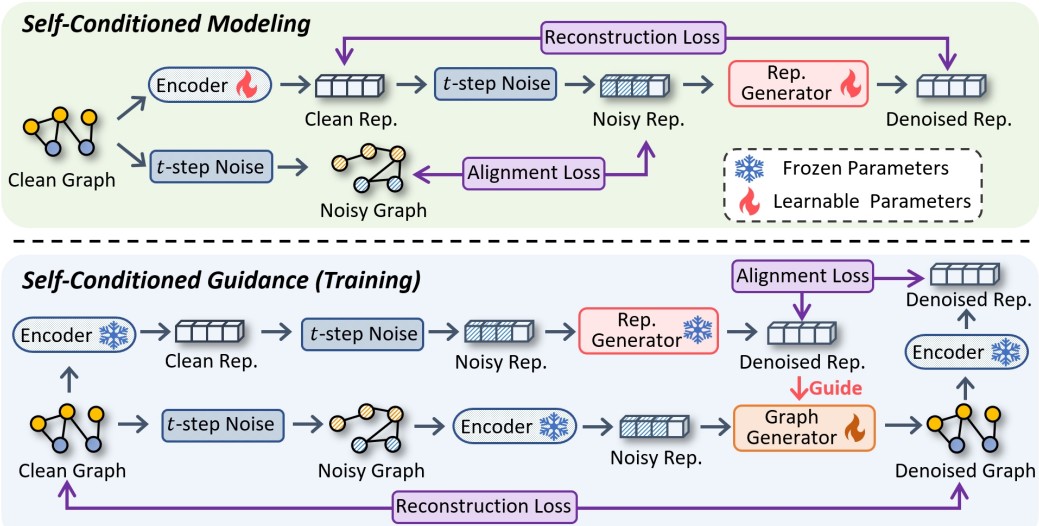

Figure 2: The overall process of GraphRCG during training. Specifically, we learn from the training data a representation generator that outputs a representation based on noise sampled from a standard Gaussian distribution. After that, a graph generator is trained to generate new graphs from noisy graphs under the guidance of the representations.

step, we perform self-conditioned guidance via bootstrapped representations with noise at the same timestep. We further enhance guidance by aligning the denoised graph with the clean representations.

In this work, we represent a graph as $G = (,)$, where $\in \mathbb{R}^{n \times a}$ and $\in \mathbb{R}^{n \times n \times b}$ contain all the one-hot encodings of nodes and edges, respectively. Here $a$ and $b$ are the numbers of node types and edge types, respectively. We consider the state of "no edge" as an edge type. $n$ is the number of nodes in $G$. Notably, although we focus on discrete categorical features (i.e., types) in this paper, our work can be easily extended to scenarios with continuous features.

## 3.1 SELF-CONDITIONED MODELING

Our framework GraphRCG aims to capture graph distributions (RQ1) by learning a low-dimensional representation distribution, such that the representations could be subsequently used for guidance during the generation process. Therefore, this requires the representation generator to comprehensively capture the complex patterns in graph datasets, in terms of both node features and structures.

To transform graphs into representations, we first employ a graph encoder $h_\eta$ (parametrized by $\eta$) to transform an input graph $G = (,)$, into a low-dimensional representation: $= h_\eta(,)$.

**Representation Generator.** Given the representations of all training data provided by the graph encoder, the representation generator is required to learn to generate representations with the same distribution. To enhance generation performance, following (Li et al., 2023), we utilize the Representation Diffusion Model (RDM) architecture, which generates representations from Gaussian noise, based on the process of Denoising Diffusion Implicit Models (DDIM) (Song et al., 2021). In particular, RDM utilizes a backbone comprising a fully connected network with multiple residual blocks. Each block is composed of an input layer, a timestep embedding projection layer, and an output layer. The number of residual blocks and the hidden dimension size both act as hyper-parameters. During training, given a representation of a graph sample, learned by the graph encoder, we first perturb it by adding random noise as follows:

$$_t = \sqrt{\alpha_t}_0 + \sqrt{1 - \alpha_t}\epsilon, \quad \text{where} \quad \epsilon \sim \mathcal{N}(\mathbf{0}, \mathbf{I}), \tag{1}$$

where $_0$ is the clean representation learned from $G$ by the graph encoder, i.e., $_0 = h_\eta(,)$. $_t$ is the noisy version of $_0$ at timestep $t$. $\alpha_{1:T} \in (0, 1]^T$ is a decreasing sequence, where $T$ is the total number of timesteps. Then the representation generator $f$, parameterized by $\gamma$, is trained to denoise the perturbed representation to obtain a clean one. In this way, the corresponding objective for the

representation generator could be formulated as follows:

$$\mathcal{L}_{RG} = \mathbb{E}_{0,\epsilon\sim\mathcal{N}(0,1),t}\left[\|0 - f_\gamma\left(t,t\right)\|_2^2\right], \tag{2}$$

where $_0$ is sampled from representations of graphs in training data, and $t$ is uniformly sampled from $\{1, 2, \ldots, T\}$. The target of the representation generator $f_\gamma$ is to capture the representation distribution and learn to generate representations from random noise. After optimization, the representation generator could perform sampling from random noise, following the DDIM strategy (Song et al., 2021), to obtain new representations.

On the other hand, the noisy representation $_t$, i.e., the input to the representation generator, is not related to any noisy graph. That being said, the noisy representation might not faithfully represent the noise added to an actual graph. Therefore, to enhance such consistency, we propose an alignment loss to train the encoder, which is formulated as follows:

$$\mathcal{L}_{AR} = \mathbb{E}_{G,t}\left[\|t - h_\eta(t,t)\|_2^2\right], \tag{3}$$

where $G_t = (t,t)$ denotes the noisy graph with the same timestep $t$. The loss $\mathcal{L}_{AR}$ is designed to align the noise processes of representations and graphs, such that the noisy representation $_t$ preserves the same information as the noisy graph at the same timestep $G_t$. In the following, we detail the process of adding noise to any graph.

**Adding Noise to Graphs.** In discrete diffusion, adding noise equates to transitioning between states, that is, choosing a state based on a categorical distribution. For each timestep $t$, the probability of moving from one state to another is defined by a Markov transition matrix $\mathbf{Q}_t$, where $\mathbf{Q}_t[i, j]$ represents the likelihood of transitioning from state $i$ to state $j$. For graph generation, these states generally represent specific node types or edge types. Particularly, an edge-type state represents the absence of an edge. The process of adding noise is performed in a disentangled manner, which operates independently across nodes and edges with separate noise perturbations. A step of the noise process could be expressed as

$$q(G_t|G_{t-1}) = (t_{-1}t; t_{-1}t), \tag{4}$$

where $t$ is the timestep. Moreover, $_t$ and $_t$ represent the transition matrices for the nodes and edges at timestep $t$, respectively.

Given the Markovian nature of the noise model, the noise addition is not cumulative, as the probability $q(G_t|G_0)$ could be directly calculated from all the respective Markov transition matrices:

$$q(G_t|G_0) = (_0\prod_{i=1}^{t} {}_i0\prod_{i=1}^{t} {}_i), \tag{5}$$

where $_0 =$ and $_0 =$ denote the input node and edge types of $G$, respectively. This formulation captures the essence of the discrete diffusion process, i.e., applying independent, state-specific transitions at each timestep for both nodes and edges in graph generation. In this manner, we could obtain $_t = _0\prod_{i=1}^{t} {}_i$ and $_t = _0\prod_{i=1}^{t} {}_i$.

To specify the Markov transition matrices, previous works have explored several feasible choices. The most prevalent choices in the literature have been uniform transitions (Austin et al., 2021; Yang et al., 2023) and absorbing transitions (Chen et al., 2023; Kong et al., 2023). However, these do not contain the distribution information and thus could not benefit the capturing of graph distribution in our framework. Therefore, in our approach, we leverage the marginal transitions (Ingraham et al., 2023; Vignac et al., 2022), in which the probability of transitioning to any given state is directly related to its marginal probability observed in the dataset. In this manner, the transition matrices are modeled in a way that mirrors the natural distribution of states in graph data. As the edges are generally sparse in the graph data, the probability of jumping to the state of "no edge" is significantly higher than that of other states. To present the noise process, we first define $\mathbf{p} \in \mathbb{R}^a$ and $\mathbf{p} \in \mathbb{R}^b$ as the marginal distributions for the node and edge types, respectively. The marginal transition matrices for nodes and edges are formulated as

$$\overline{}_t = \alpha^t\mathbf{I} + \beta^t\mathbf{1}_a(\mathbf{p})^\top, \quad \text{and} \quad _t = \alpha^t\mathbf{I} + \beta^t\mathbf{1}_b(\mathbf{p})^\top. \tag{6}$$

The above formulation ensures that $\lim_{t\to\infty}\prod_{i=1}^{t}\overline{}_i = \mathbf{P}$, where each column in $\mathbf{P}$ is $\mathbf{p}$. The equation also holds true for the edges. More details of the noise model are provided in Appendix A. The overall process of our self-conditioned modeling module is presented in Algorithm 1.

Figure 3: The sampling process of our framework with self-conditioned guidance. Our step-wise incorporation strategy employs the denoised presentation at each timestep to guide the denoising of the noisy graph at the same time step, thereby progressively guiding each step closer to the learned distributions.

## 3.2 SELF-CONDITIONED GUIDANCE (TRAINING)

In self-conditioned guidance for RQ2, we optimize a graph generator to create new graphs conditioned on bootstrapped representations from the representation generator. Existing works have explored various methods to perform generation conditioned on specific properties, such as high activity in molecular graph generation (Huang et al., 2022). However, these properties are generally several scalar values, which could capture only a small fraction of the information in the dataset. Moreover, these approaches often adopt a classifier or regressor to guide the generation process, which could not exploit the useful information in representations. In contrast, using representations as guidance for generation could largely benefit from the learned distribution in our representation generator.

As our representation generator is implemented by a diffusion model, we aim to utilize the information from not only the generated representation but also its generation process. With this in mind, we implement the generator as a denoising diffusion model, so that each step of diffusion could benefit from the representation of the same denoising step in a boostrapped manner. Our graph generator $g_\theta$, implemented as a denoising network and parametrized by $\theta$, is trained to predict the clean graph, given a noisy graph $G_t = (_{t}, _t)$ at a randomly sampled timestep $t$:

$$(\widetilde{\mathbf{p}}, \widetilde{\mathbf{p}}) = g_\theta(_{t}, \widetilde{_t}, t),\tag{7}$$

where $\widetilde{_t} = f_\gamma(_t, t)$ is the denoised representation, generated from the representation generator at timestep $t$. Moreover, $\widetilde{\mathbf{p}} \in \mathbb{R}^a$ and $\widetilde{\mathbf{p}} \in \mathbb{R}^b$ are the predicted distributions for the node types and the edge types, respectively. The graph generator is based on the message-passing transformer architecture (Shi et al., 2020), as it effectively extracts the complex correlations between nodes and edges, while also being suitable for incorporating representations for conditioning (Vignac et al., 2022). Specifically, the layers incorporate the graph attention mechanism (Veličković et al., 2018) into a Transformer framework (Vaswani et al., 2017), achieved by including normalization and feedforward layers. For each given (noisy) graph, the graph generator projects nodes and edges separately into low-dimensional representations and processes them through totally $L$ transformer layers, denoted as $(_{t}^{(l+1)}, _{t}^{(l+1)}) = M_{(l)}(_{t}^{(l+1)}, _{t}^{(l+1)})$. Here $_t^{(l)}$ (or $_t^{(l)}$) is the representation for the nodes (or edges) in the $l$-th layer of the transformer, denoted as $M_{(l)}$.

**Cross-Attention.** To effectively utilize the representations for guidance, we map them to the intermediate layers of the graph transformer via the cross-attention mechanism (Vaswani et al., 2017), achieved by

$$\text{Attention}(Q, K, V) = \text{softmax}(QK^T/\sqrt{d}) \cdot V.\tag{8}$$

Specifically, the values of $Q$, $K$, and $V$ are computed as follows:

$$Q = W_Q^{(l)} \cdot ^{(l)}, K = W_K^{(l)} \cdot \widetilde{_t}, V = W_V^{(l+1)} \cdot \widetilde{_t},\tag{9}$$

where $W_Q^{(l)} \in \mathbb{R}^{d \times d_x}, W_K^{(l)} \in \mathbb{R}^{d \times d_h}$, $W_V^{(l)} \in \mathbb{R}^{d \times d_h}$ are learneable projection matrices. $d_x$ and $d_h$ are the dimensions of and , respectively. Notably, the above process is performed for edge representations $^{(l)}$ in the same way with different parameters.

With the representations as guidance, the denoising network is then tasked to predict the clean graph, given the noisy graph $G_t = (_{t}, _t)$. The reconstruction objective is described as follows:

$$\mathcal{L}_{GG} = \mathbb{E}_{G,t}\left[\sum_{i=1}^n \text{CE}(_i, \widetilde{\mathbf{p}}^i) + \sum_{i=1}^n \sum_{j=1}^n \text{CE}(_{i,j}, \widetilde{\mathbf{p}}^{i,j})\right],\tag{10}$$

where $t$ is uniformly sampled from $\{1, 2, \ldots, T\}$. CE$(\cdot, \cdot)$ denotes the cross-entropy loss, as the prediction results of  and  are categorical, obtained from Eq. (7).

In addition, as we expect the representation to guide graph generation, we also aim to align the representation of the generated graph with the clean representation. In particular, we introduce another alignment loss that aligns the generated (clean) graph with the denoised (clean) representation from the representation generator. The loss is formulated as follows:

$$\mathcal{L}_{AG} = \mathbb{E}_{G,t}\left[\left\|h_\eta(p^{,}p^{)} - f_\gamma\left(_t, t\right)\right\|_2^2\right]. \tag{11}$$

Note that $\mathcal{L}_{AG}$, together with $\mathcal{L}_{GG}$, will be only used for optimizing the graph generator, not involving the representation generator. In this way, we can ensure that the representation generator focuses on capturing the graph distribution. The detailed overall process of self-conditioned guidance is presented in Algorithm 2.

### 3.3 Self-Conditioned Guidance (Sampling)

After optimization, our graph generator could be used to create new graphs. Specifically, we first sample a fixed number of nodes $n$ based on the prior distribution of the graph size in the training data, and $n$ remains fixed during generation. Next, a random graph is sampled from the prior graph distribution $G_T \sim \mathbf{p}^\times \mathbf{p}$, where $\mathbf{p}$ and $\mathbf{p}$ represent the marginal distribution for each node type and edge type present in the dataset, respectively. Note that $\mathbf{p}$ and $\mathbf{p}$ are both categorical distributions. As presented in Fig. 3, with the sampled noisy graph $G_T$, we could leverage the generator to recursively sample a cleaner graph $G_{t-1}$ from the previous graph $G_t$. As we perform self-conditioned guidance, the sampling process within each step should also involve representations.

**Step-wise Incorporation of Bootstrapped Represenations.** We introduce guidance into each sampling step with the representation obtained at the same timestep. That being said, for the representation generator, we utilize all (intermediate) representations during sampling, instead of only the last clean one. In this manner, the sampling process is described as

$$G_{t-1} \sim p_\theta(G_{t-1}|G_{t,t-1}), \quad \text{where} \quad _{t-1} = f_\gamma(_t, t). \tag{12}$$

In concrete, we keep track of the representation sampling process in the representation generator and utilize the representation in each step to guide the graph sampling in the same timestep $t$. In this manner, the generation process will absorb the graph distribution information learned by the representation generator, thereby improving generation performance.

## 4 Experiments

In our experiments, we evaluate GraphRCG across graph datasets covering realistic molecular and synthetic non-molecular datasets, in comparison to baselines, including auto-regressive models: GRAN (Liao et al., 2019) and GraphRNN (You et al., 2018b), a GAN-based model: SPECTRE (Martinkus et al., 2022), and diffusion models: EDP-GNN (Niu et al., 2020), DiGress (Vignac et al., 2022), GDSS (Jo et al., 2022), GraphARM (Kong et al., 2023), HiGen (Karami, 2023), and SparseDiff (Qin et al., 2023). We provide implementation details and hyperparameter settings in Appendix D.2.

For generic graph generation, we evaluate the quality of the generated graphs with structure-based evaluation metrics. We follow previous work (You et al., 2018b) and calculate the MMD (Maximum Mean Discrepancy) between the graphs in the test set and the generated graphs, regarding (1) degree distributions, (2) clustering coefficients distributions, (3) the number of orbits with four nodes, and (4) the spectra of the graphs obtained from the eigenvalues of the normalized graph Laplacian (You et al., 2018b; Chen et al., 2023). For molecular graph generation, following previous works (Jo et al., 2022; Kong et al., 2023), we evaluate the generated molecular graphs with several key metrics: (1) Frechet ChemNet Distance (FCD) (Preuer et al., 2018), (2) Neighborhood Subgraph Pairwise Distance Kernel (NSPDK) MMD (Costa & De Grave, 2010), (3) Validity, and (4) Uniqueness. We provide details of these metrics in Appendix D.3. Our code is provided at https://anonymous.4open.science/r/GraphRCG-D304/README.md.

Table 1: Comparison of generation results on SBM, Planar, and Ego. The best results are shown in **bold**.

| Model | SBM | | | | Planar | | | | Ego | | | |
|---|---|---|---|---|---|---|---|---|---|---|---|---|
| | Deg. ↓ | Clus. ↓ | Orbit ↓ | Spec. ↓ | Deg. ↓ | Clus. ↓ | Orbit ↓ | Spec. ↓ | Deg. ↓ | Clus. ↓ | Orbit ↓ | Spec. ↓ |
| Training | 0.0008 | 0.0332 | 0.0255 | 0.0063 | 0.0002 | 0.0310 | 0.0005 | 0.0052 | 0.0002 | 0.0100 | 0.0120 | 0.0014 |
| GraphRNN | 0.0055 | 0.0584 | 0.0785 | 0.0065 | 0.0049 | 0.2779 | 1.2543 | 0.0459 | 0.0768 | 1.1456 | 0.1087 | - |
| GRAN | 0.0113 | 0.0553 | 0.0540 | 0.0054 | 0.0007 | 0.0426 | **0.0009** | 0.0075 | 0.5778 | 0.3360 | 0.0406 | - |
| SPECTRE | 0.0015 | 0.0521 | 0.0412 | 0.0056 | 0.0005 | 0.0785 | 0.0012 | 0.0112 | - | - | - | - |
| DiGress | 0.0013 | 0.0498 | 0.0433 | - | 0.00027 | 0.0563 | 0.0098 | 0.0062 | 0.0708 | 0.0092 | 0.1205 | - |
| HiGen | 0.0019 | 0.0498 | 0.0352 | 0.0046 | - | - | - | - | 0.0472 | **0.0031** | 0.0387 | 0.0062 |
| SparseDiff | 0.0016 | 0.0497 | **0.0346** | 0.0043 | 0.0007 | 0.0447 | 0.0017 | 0.0068 | 0.0019 | 0.0537 | 0.0209 | 0.0050 |
| GraphRCG | **0.0011** | **0.0475** | 0.0378 | **0.0038** | **0.00025** | **0.0341** | 0.0010 | **0.0059** | **0.0015** | 0.0448 | **0.0183** | **0.0042** |

Table 2: Results of various methods on the QM9 and ZINC250k Datasets. The best results are shown in **bold**.

| Model | QM9 Dataset | | | | ZINC250k Dataset | | | |
|---|---|---|---|---|---|---|---|---|
| | Validity↑ | NSPDK↓ | FCD↓ | Unique↑ | Validity↑ | NSPDK↓ | FCD↓ | Unique↑ |
| EDP-GNN | 47.52 | 0.005 | 2.68 | **99.25** | 82.97 | 0.049 | 16.74 | **99.79** |
| SPECTRE | 87.33 | 0.163 | 47.96 | 35.7 | 90.20 | 0.109 | 18.44 | 67.05 |
| GDSS | 95.72 | 0.003 | 2.9 | 98.46 | **97.01** | **0.019** | 14.66 | 99.64 |
| DiGress | **99.01** | **0.0005** | 0.36 | 96.66 | 91.02 | 0.082 | 23.06 | 81.23 |
| GraphARM | 90.25 | 0.002 | 1.22 | 95.62 | 88.23 | 0.055 | 16.26 | 99.46 |
| GraphRCG | **99.12** | 0.0008 | **0.28** | 98.39 | 92.38 | 0.041 | **13.48** | 96.15 |

## 4.1 COMPARATIVE RESULTS

**Generic Graph Generation.** In this subsection, we further evaluate our framework on generic graph datasets with relatively larger sizes than molecular graphs. In particular, we consider two synthetic datasets: SBM, drawn from stochastic block models (Martinkus et al., 2022), with a maximum size of 200 nodes, and Planar, containing planar graphs with a fixed size of 64 (Vignac et al., 2022). In addition, we consider a realistic citation dataset Ego (Sen et al., 2008), originated from Citeseer (Giles et al., 1998). Further details of these datasets are provided in Appendix D.1. From the results presented in Table 1, we could obtain the following observations: (1) GraphRCG outperforms other baselines on all three datasets across various metrics for graph generation, demonstrating the effectiveness of our framework in precisely capturing graph distributions and utilizing them for generation guidance. (2) The performance improvement over other methods is more substantial on the Planar dataset. As illustrated in the t-SNE plot in Fig. 7 (b), the distribution of the Planar dataset is more scattered, increasing the difficulty of accurately capturing it. Nevertheless, our framework learns graph distributions with a representation generator, which enables the modeling of complex underlying patterns. (3) GraphRCG is particularly competitive in the MMD score regarding degree distributions and the number of orbits. This observation demonstrates that GraphRCG could authentically capture the complex graph distribution of the training samples with the help of self-conditioned modeling. We include additional visualization results of graphs generated by our framework in Appendix E.

**Molecular Graph Generation.** To evaluate our framework on molecular graph generation, we select two popular datasets: QM9 (Wu et al., 2018) and ZINC250k (Irwin et al., 2012), with details and provided in Appendix D.1. We present the molecular graph generation results on QM9 in Table 2. Specifically, GraphRCG demonstrates competitive performance on QM9 across various metrics, compared to other state-of-the-art baselines. The best FCD values on QM9 and ZINC250k indicate that GraphRCG effectively captures the chemical property distributions in the dataset. Furthermore, the outstanding validity score on QM9 also signifies that our framework GraphRCG is capable of generating valid molecules that are more closely aligned with the training data distribution.

## 4.2 REPRESENTATION INTERPOLATION

As our self-conditioned guidance leverages representations, we could manually perform linear interpolation for two representations to generate graphs that represent the properties of both representations. With our step-wise incorporation strategy, we extract two series of representations generated by our

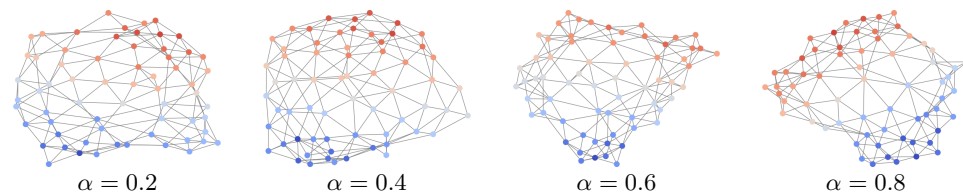

$\alpha = 0.2$      $\alpha = 0.4$      $\alpha = 0.6$      $\alpha = 0.8$

Figure 4: The generated graphs from Planar with different interpolation ratios between two representations.

representation generator for all timesteps, i.e., $t = 1, 2, \ldots, T$, and perform interpolation for each timestep. We denote $\alpha$ as the interpolation ratio, where $\alpha = 0$ and $\alpha = 1$ indicate that we entirely utilize one of the representations. We provide the visualization results in Fig. 4, from which we observe that the generated graphs guided by interpolated representations remain meaningful with different interpolation ratios. This result demonstrates that our representation generator is capable of capturing smooth graph distributions with rich information. Furthermore, our design also enables further applications with specific representations as guidance.

## 4.3 ABLATION STUDY

**Effect of Self-Conditioned Modeling.** Previous works (Vignac et al., 2022) have also explored various strategies to capture and model distributions for the guidance of graph generation. For example, DiGress leverages manually designed structural features (e.g., the number of cycles) for graph generation. However, such features contain less information and require domain knowledge. In this subsection, we compare several variants of our framework with these structural features to evaluate the efficacy of self-conditioned modeling in encapsulating the graph distribution. We consider using the following features (representations) to replace our self-conditioned modeling module: (1) structural features proposed in DiGress, (2) representations of true training samples, which limit the uniqueness of generation, (3) a mixture of true representations, and (4) distributions learned by a Gaussian Mixture Model (GMM). As the structural features involve molecular information, we conduct experiments on the QM9 dataset with explicit hydrogens, which is a more complex setting with larger graphs.

From the results presented in Table 3, we first observe that the inclusion of structural features significantly enhances the performance of DiGress, especially in the metric of molecule stability, which benefits from the integration of molecular characteristics. The substitution of self-conditioned modeling with structural features, however, results in a notable decline in performance, highlighting the importance of distributions with comprehensive information for effective guidance. Furthermore, among the variants that directly operate on training sample representations, it is evident that

Table 3: The ablation study results regarding self-conditioned modeling on dataset QM9 with explicit hydrogen.

| Model | Valid↑ | Unique↑ | Atom S.↑ | Mol S.↑ |
|---|---|---|---|---|
| Dataset | 97.8 | 100 | 98.5 | 87.0 |
| DiGress w/o A | 92.3 | 97.9 | 97.3 | 66.8 |
| DiGress w/ A | 95.4 | 97.6 | **98.1** | 79.8 |
| GraphRCG w/ A | 92.9 | 95.4 | 93.1 | 76.6 |
| GraphRCG w/ T | 91.2 | 91.5 | 90.5 | 72.3 |
| GraphRCG w/ T+M | 94.6 | 97.0 | 90.2 | 74.9 |
| GraphRCG w/ GMM | 96.4 | 94.1 | 91.5 | 77.1 |
| GraphRCG | **96.9** | **98.1** | 97.2 | **81.9** |

the mere utilization of pre-existing representations yields suboptimal outcomes, especially concerning the metric of uniqueness. This suggests that relying solely on existing representations captures only a small portion of the authentic distribution, thereby adversely impacting the overall performance.

**Effect of Self-Conditioned Guidance.** In this subsection, we investigate the effect of self-conditioned guidance in our framework. We first replace the entire guidance strategy with a gradient-based method, which leverages computed gradients as guidance. Therefore, the graph generator does not involve any representation during training. For the second variant, we remove the alignment loss $\mathcal{L}_{AG}$ described in Eq. (11). Without this loss, the graph generator is not well-aligned with the representations produced by the representation generator, thereby affecting the guidance performance. For the third variant, we directly use the fixed representation, i.e., the clean representation, to guide generation. In this case, the step-wise guidance strategy is removed, resulting in the infeasibility of progressive

Table 4: The ablation study results on different variants of our framework GraphRCG regarding self-conditioned guidance on the dataset Ego.

| Dataset | Ego | | | |
|---|---|---|---|---|
| Model | Deg. ↓ | Clus. ↓ | Orbit ↓ | Spec. ↓ |
| GraphRCG-Gradient | 0.0134 | 0.0955 | 0.0464 | 0.0151 |
| GraphRCG w/o $\mathcal{L}_{AG}$ | 0.0088 | 0.0647 | 0.0252 | 0.0092 |
| GraphRCG-fixed | 0.0053 | 0.0653 | 0.0310 | 0.0125 |
| GraphRCG | **0.0015** | **0.0448** | **0.0183** | **0.0042** |

guidance. We provide further details regarding these variants in Appendix D.5. From the results presented in Table 4, we could first observe that our framework outperforms all other variants in most evaluation metrics, demonstrating the effectiveness of our self-conditioned guidance strategy. Moreover, GraphRCG-Gradient and GraphRCG w/o $\mathcal{L}_{AG}$ exhibit significantly poorer performance, as evidenced by higher scores across all metrics. This deterioration, particularly in Clus. and Orbit values, underscores the difficulty in preserving graph distributions without representation guidance or alignment loss. GraphRCG-fixed shows improved performance relative to the other variants, demonstrating the benefits of representation guidance even with a fixed one. Nevertheless, the results also indicate that the step-wise self-conditioned guidance is more beneficial for progressively guiding the generation process.

Table 5: The performance of various methods on three generic datasets.

| Model | Community | | | Cora | | | Enzymes | | |
|---|---|---|---|---|---|---|---|---|---|
| | Deg. ↓ | Clus. ↓ | Orbit ↓ | Deg. ↓ | Clus. ↓ | Orbit ↓ | Deg. ↓ | Clus. ↓ | Orbit ↓ |
| SPECTRE | 0.048 | 0.049 | 0.016 | 0.021 | 0.080 | 0.007 | 0.136 | 0.195 | 0.125 |
| GDSS | 0.045 | 0.086 | 0.007 | 0.160 | 0.376 | 0.187 | 0.026 | 0.061 | 0.009 |
| DiGress | 0.047 | **0.041** | 0.026 | 0.044 | 0.042 | 0.223 | **0.004** | 0.083 | **0.002** |
| GraphARM | **0.034** | 0.082 | **0.004** | 0.273 | 0.138 | **0.105** | 0.029 | 0.054 | 0.015 |
| GraphRCG | 0.040 | 0.053 | 0.029 | **0.038** | **0.036** | 0.173 | **0.004** | **0.079** | **0.002** |

## 4.4 RESULTS ON ADDITIONAL GENERIC DATASETS

In this subsection, we consider additional experiments on three generic datasets. As our framework is based on discrete graph diffusion, we mainly follow the dataset settings used in DiGress (Vignac et al., 2022) and the following work SparseDiff (Qin et al., 2023) for the three generic datasets. We consider the following datasets: Community (You et al., 2018b), Cora (Sen et al., 2008), and Enzymes (Schomburg et al., 2004), order to further improve the integrity of our evaluation. We provide the results in Table 5. The results of other baselines are obtained from GraphARM (Kong et al., 2023). From the results, we could observe that our framework also achieves competitive results, compared to other methods on various datasets. The performance is particularly better on dataset Cora and Enzymes, indicating that our framework is generalizable to various datasets.

## 5 CONCLUSION

In this work, we investigate the importance of capturing and utilizing distributions for graph generation. We propose a novel self-conditioned generation, that encompasses self-conditioned modeling and self-conditioned guidance. Instead of directly learning from graph distributions, we encode all graphs into representations to capture graph distributions, which could preserve richer information. Our self-conditioned guidance module further guides the generation process in each timestep with representations with different degrees of noise. We conduct extensive experiments to evaluate our framework, and the results demonstrate the efficacy of our framework in graph generation.

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

## A  Noise Model

In this section, we provide additional information on the noise model used during the training of our graph generator. Particularly, the graphs at timestep $t$ could be represented as $_t =_0 \prod_{i=1}^{t} {}_i$ and $_t =_0 \prod_{i=1}^{t} {}_i$. Here, $_i$ and $_i$ are defined in Eq. (13), i.e.,

$$\overline{\overline{}}_t \alpha^t \mathbf{I} + \beta^t \mathbf{1}_a(\mathbf{p})^\top, \;\; \text{and} \;\; _t = \alpha^t \mathbf{I} + \beta^t \mathbf{1}_b(\mathbf{p})^\top. \tag{13}$$

In practice, the noise process is not cumulative. By defining $\overline{\alpha}^t = \prod_{i=1}^{t} \alpha^i$ and $\overline{\beta}^t = 1 - \overline{\alpha}^t$, we could achieve

$$\prod_{i=1}^{t} \overline{\overline{}}_i \overline{\alpha}^t \mathbf{I} + \overline{\beta}^t \mathbf{1}_a(\mathbf{p})^\top, \;\; \text{and} \;\; \prod_{i=1}^{t} \overline{\overline{}}_i \overline{\alpha}^t \mathbf{I} + \overline{\beta}^t \mathbf{1}_a(\mathbf{p})^\top. \tag{14}$$

Following (Vignac et al., 2022), we set $\overline{\alpha}^t = \cos(0.5\pi(t/T + s)/(1 + s))^2$ with $s$ being a small value. In this manner, when computing the noisy graphs $G_t = (_{t}, _t)$, we do not need to recursively multiply $Q_t$. Instead, we utilize the value of $\overline{\alpha}^t$ to directly obtain $G_t$.

The noise model for the representation generator is similar and in a simpler form. We first sample $\epsilon \sim \mathcal{N}(0, \mathbf{I})$ and compute the noisy representations based on $_t = \sqrt{\alpha_t} {}_0 + \sqrt{1 - \alpha_t} \epsilon$.

## B  Self-conditioned Modeling

In this section, we present the detailed algorithm of our self-conditioned modeling module in Algorithm 1. Specifically, we aim to optimize a representation generator along with a graph encoder. The overall process is presented in Fig. 2.

---

**Algorithm 1** The training process of self-conditioned modeling.

---

**Require:** A training graph distribution $\mathcal{G}$, maxminum number of denoising steps $T$, number of training epochs $T_{tr}$ .
**Ensure:** Optimized graph encoder and representation generator.
1: **for** $i = 1, 2, \ldots, T_{tr}$ **do**
2:     Sample $G = (,)$ from $\mathcal{G}$;
3:     Sample $t \sim \mathcal{U}(1, 2, \ldots, T)$ and $\epsilon \sim \mathcal{N}(0, \mathbf{I})$;
       // Training the representation generator
4:     $_0 \leftarrow h_\eta(,)$;
5:     $_t \leftarrow \sqrt{\alpha_t} {}_0 + \sqrt{1 - \alpha_t} \epsilon$;
6:     $\widetilde{}_t = f_\gamma(_t, t)$;
7:     Optimize $f_\gamma$ with $\mathcal{L}_{RG}$ in Eq. (2); // Training the graph encoder
8:     Sample $G_t = (_{t}, _t)$ from $\prod_{i=1}^{t} {}_i^\times \prod_{i=1}^{t} {}_i$;
9:     $_t = h_\eta(_{t}, _t)$;
10:    Optimize $h_\eta$ with $\mathcal{L}_{AR}$ in Eq. (3);
11: **end for**

---

## C  Self-conditioned Guidance

### C.1  Comparison with Other Methods

In this subsection, we discuss and compare our framework GraphRCG with other diffusion-based generative models for graph data. We provide the comparisons regarding technical details in Table 6. Typically, existing diffusing methods for graph generation rely on either continuous or discrete diffusion. As a classic example, GDSS (Jo et al., 2022) adopts Gaussian transition kernels to perform continuous diffusion with a score-matching strategy. DiGress, on the other hand, performs discrete diffusion while considering categorical features of nodes and edges. Nevertheless, although continuous representations could capture complex structural patterns in graph distributions, they are less effective in generating discrete graph data. In contrast, our framework is capable of leveraging

---

**Algorithm 2** Detailed training and sampling process of our self-conditioned guidance module.

---

**Require:** A training graph distribution $\mathcal{G}$, maxminum number of denoising steps $T$, number of training epochs $T_{tr}$ .

**Ensure:** A generated graph that aligns with the distribution $\mathcal{G}$.

    `// Training phase`

1: **for** $i = 1, 2, \ldots, T_{tr}$ **do**

2:    Sample $G = (,)$ from $\mathcal{G}$;

3:    Sample $t \sim \mathcal{U}(1, 2, \ldots, T)$ and $\epsilon \sim \mathcal{N}(0, \mathbf{I})$;

4:    Sample $G_t = (_{t,t})$ from $\prod_{i=1}^{t} {}_i \times \prod_{i=1}^{t} {}_i$;

5:    $_0 \leftarrow h_\eta(,)$ and $_t \leftarrow \sqrt{\alpha_t}_0 + \sqrt{1 - \alpha_t}\epsilon$;

6:    $(\widetilde{\mathbf{p}}, \widetilde{\mathbf{p}}) \leftarrow g_\theta(_{t,t}, \widetilde{,t}, t)$;

7:    $\widetilde{}_t = f_\gamma(_t, t)$;

8:    Optimize $g_\theta$ with $\mathcal{L}_{GG}$ in Eq. (10) and $\mathcal{L}_{AG}$ in Eq. (11);

9: **end for**

    `// Sampling phase`

10: Sample $n$ from the training data distribution of graph sizes;

11: Sample $G_T = (_{T,T}) \sim \mathbf{p}^\times \mathbf{p}$;

12: **for** $t = T, T-1, \ldots, 1$ **do**

13:    $_t = h_\eta(_{T,T})$;

14:    $_{t-1} = f_\gamma(_t, t)$;

15:    $(\widetilde{\mathbf{p}}, \widetilde{\mathbf{p}}) \leftarrow g_\theta(_{t,t}, _{t-1}, t)$;

16:    $G_{t-1} \sim p_\theta(G_{t-1}|G_{t,t-1}) = \prod_{i=1}^{n} \widetilde{\mathbf{p}}_i \prod_{i=1}^{n} \prod_{j=1}^{n} \widetilde{\mathbf{p}}^{i,j}$;

17: **end for**

18: **return** $G_0$.

---

Table 6: Comparison of different denoising diffusion models for graph generation. $G(n, p)$ denotes the Erdős-Rényi graph model (Erdős et al., 1960), where $p$ is the probability of an edge existing between two nodes, and $n$ is the graph size.

| Model | Diffusion Type | Convergence | Conditional Generation | Featured Generation |
|---|---|---|---|---|
| EDP-GNN | Continous | $\mathcal{N}(0, 1)$ | - | - |
| GDSS | Continous | $\mathcal{N}(0, 1)$ | - | ✓ |
| DiscDDPM | Discrete | $G(n, 0.5)$ | - | - |
| DiGress | Discrete | Empirical | Gradients from a classifier | ✓ |
| SparseDiff | Discrete | Empirical | - | ✓ |
| EDGE | Discrete | $G(n, 0)$ | Degree sequence | ✓ |
| Ours | Continous & Discrete | $\mathcal{N}(0, 1)$ & Empirical | Representation | ✓ |

continuous diffusion to guide discrete diffusion, thereby combining the strengths of both and allowing for the generation of a wider range of graph structures. Regarding the convergence type, which refers to the pure noise state, we also combine both continuous and discrete noise to facilitate the optimization of both our representation generator and graph generator. Comparing the conditional generation type, our framework is conditioned on representations, which preserve richer information of complex distributions of each dataset. As a result, the generation process could significantly benefit from the guidance of representations.

## D  EXPERIMENTAL SETTINGS

Table 7: Detaile statistics of generic datasets used in our experiments.

| Name | Graph number | Node range | Edge range |
|---|---|---|---|
| Planar | 200 | [64, 64] | [346, 362] |
| SBM | 200 | [44, 187] | [258, 2258] |
| Ego | 757 | [50, 399] | [112, 2124] |

### D.1 DATASET DETAILS

For generic datasets, we consider SBM, Planar, and Ego. In particular, the two synthetic datasets, SBM and Planar, are obtained following the settings in SPECTRE (Martinkus et al., 2022). The Ego dataset setting follows SparseDiff (Qin et al., 2023).

- The SBM dataset contains 2022 Stochastic Block Model graphs. Each graph has 2 to 5 communities, each of which has 20 to 40 nodes. The inter-community edge probability is 0.3 and the intra-community edge probability is 0.05.
- The Planar dataset contains 200 planar graphs, each with 64 nodes. The graphs are generated via Delaunay triangulation on a set of randomly placed points.
- The Ego dataset contains 757 graphs, each with 50 399 nodes, sampled from the Citeseer Network Dataset (Giles et al., 1998).

The detailed statistics of these datasets are provided in Table 7.

For molecular datasets, we consider ZINC250k and QM9 following GraphARM (Kong et al., 2023). Particularly, QM9 consists of molecular graphs with a maximum of nine heavy atoms that could be treated with implicit or explicit hydrogens. We provide the details of dataset statistics in Table 8. ZINC-250k consists of approximately 250,000 drug-like molecules, each of which has up to 38 atoms. The dataset contains nine atom types and three edge types.

Table 8: Detaile statistics of molecular datasets used in our experiments.

| Dataset | Number of graphs | Node range | Number of node types | Number of edge types |
| --- | --- | --- | --- | --- |
| QM9 | 133,885 | [1, 9] | 4 | 3 |
| ZINC250k | 249,455 | [6, 38] | 9 | 3 |

### D.2 TRAINING SETTINGS

All experiments in our evaluation part are conducted on an NVIDIA A6000 GPU with 48GB of memory. For the specific parameters in each dataset, we follow the settings in DiGress (Vignac et al., 2022) and SparseDiff (Qin et al., 2023). We utilize the Adam optimizer (Kingma & Ba, 2015) for optimization. We also utilize the Xavier initialization (Glorot & Bengio, 2010). For the representation generator, we implement it as an RDM (Representation Diffusion Model) used in RCG (Li et al., 2023). The representation dimension size is set as 256 for all datasets. The learning rate of the RDM is set as $10^4$, and the weight decay rate is set as 0.01. The number of residual blocks in the RDM is set as 12. The batch size is various across datasets, adjusted according to the GPU memory consumption in practice. The total number of timesteps $T$ is set as 1,000. For the graph transformer architecture used to implement our graph generator, we set the number of layers as 8, with the hidden dimension size as 256. We provide our code in the supplementary materials.

### D.3 EVALUATION METRICS

For molecular graph generation, following previous works (Jo et al., 2022; Kong et al., 2023), we evaluate the generated molecular graphs with several key metrics: (1) Frechet ChemNet Distance (FCD) (Preuer et al., 2018), which quantifies the discrepancy between the distributions of training and generated graphs, based on the activation values obtained from the penultimate layer in ChemNet. (2) Neighborhood Subgraph Pairwise Distance Kernel (NSPDK) MMD (Costa & De Grave, 2010), which measures the Maximum Mean Discrepancy (MMD) between the generated and the test molecules, considering both node and edge features. (3) Validity, which refers to the proportion of generated molecules that are structurally valid without requiring valency adjustments. (4) Uniqueness, which measures the proportion of unique molecules that are distinct from each other.

### D.4 REQUIRED PACKAGES

We list the required packages for running the experiments below.

- Python == 3.9.18
- torch == 2.0.1
- pytorch_lightning==2.0.4
- cuda == 11.6
- scikit-learn == 1.3.2
- pandas==1.4.0
- torch_geometric==2.3.1
- torchmetrics==0.11.4
- torchvision==0.15.2+cu118
- numpy == 1.23.0
- scipy == 1.11.0
- wandb==0.15.4
- tensorboard == 2.15.1
- networkx == 2.8.7

### D.5 ABLATION STUDY FOR SELF-CONDITIONED GUIDANCE

With different ways to incorporate the bootstrapped representations, in our ablation study, we explore the following possible sampling strategies with self-conditioned guidance.

**Gradient-based Guidance.** In this method, the incorporation of representations is not explicitly performed in graph transformer layers. Instead, the representation guidance is performed via a gradient-based strategy, where each sampling step will output graphs that are closer to the given representation.

$$G_{t-1} \sim p_\theta(G_{t-1}|G_t)p_\eta(_t|G_{t-1}), \tag{15}$$

where $p_\eta(_t|G_{t-1}) \propto \exp(-\lambda \langle \nabla_{G_t}\|_t - h_\eta(G_t)\|_2^2, G^{t-1}\rangle)$. The form is based on the classifier-guided DDIM sampling strategy in ADM (Dhariwal & Nichol, 2021). As a result, the training of the denoising network does not involve representation, which reduces the complexity during optimization. However, as the optimizations of the representation generator and the graph generator are separate, the guidance effect of representations cannot be guaranteed during sampling.

**Guidance with Fixed Representations.** In this method, the sampling step is based on the forward process in the denoising network. However, the representation is generated from the last sampling step in the representation generator, and is fixed during graph generation. In other words,

$$G_{t-1} \sim p_\theta(G_{t-1}|G_{t,0}). \tag{16}$$

The benefit of this strategy is that the representation used for guidance is clean, and maximally contains the knowledge learned in the representation generator.

## E VISUALIZATION RESULTS

In this section, we provide additional visualization results from the experiments of generic graph generation. Particularly, we sample different generation results from the SBM and Planar datasets. The results are provided in Fig. 5 and Fig. 6.

## F REPRESENTATION QUALITY

In this section, we provide visualization results to evaluate the quality of the generated representations. We present the t-SNE plot of representations from training samples and our representation generator on datasets SBM and Planar in Fig. 7. From the visualization results, we could observe that our representation generator could faithfully capture the graph distributions from training samples. In other words, the generated representations closely align with training graph distributions. More importantly, our framework can generate representations that slightly deviate from the training

samples in specific directions. This indicates that our framework is capable of discovering novel graph distributions that are not present in training samples, while still ensuring the validity of these representations for further guidance during generations.

## G LIMITATION

In this section, we discuss the potential limitations presented in our framework. Although our framework is capable of generating graphs conditioned on the representations provided by our representation generator, it also means that the optimization quality of the representation generator is critical for graph generation. In other words, if the representation generator is not well-trained, the output representations could be detrimental to graph generation. In addition, the learned representations are not evaluated for their quality, as there are not ground truths for evaluation. A feasible

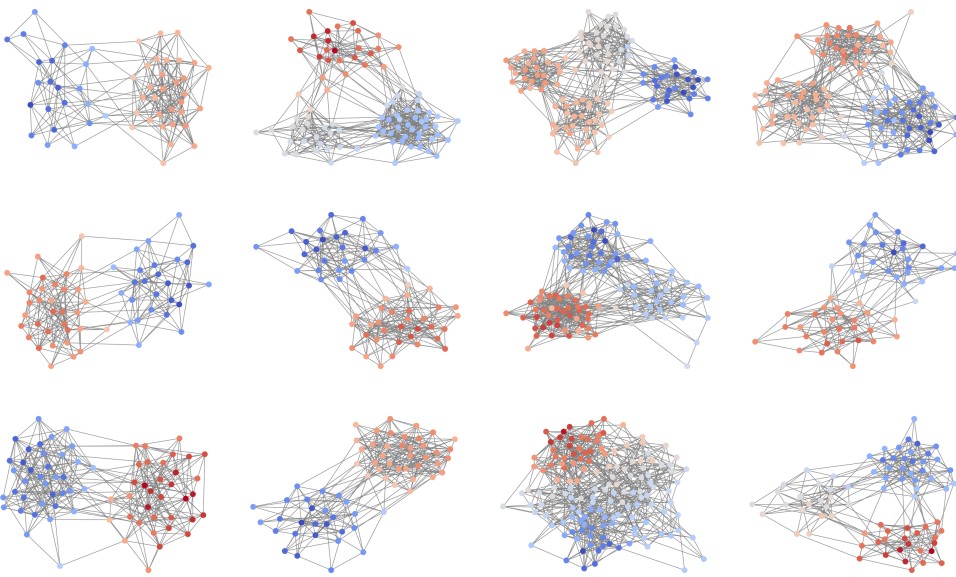

Figure 5: The generated graphs from the SBM dataset.

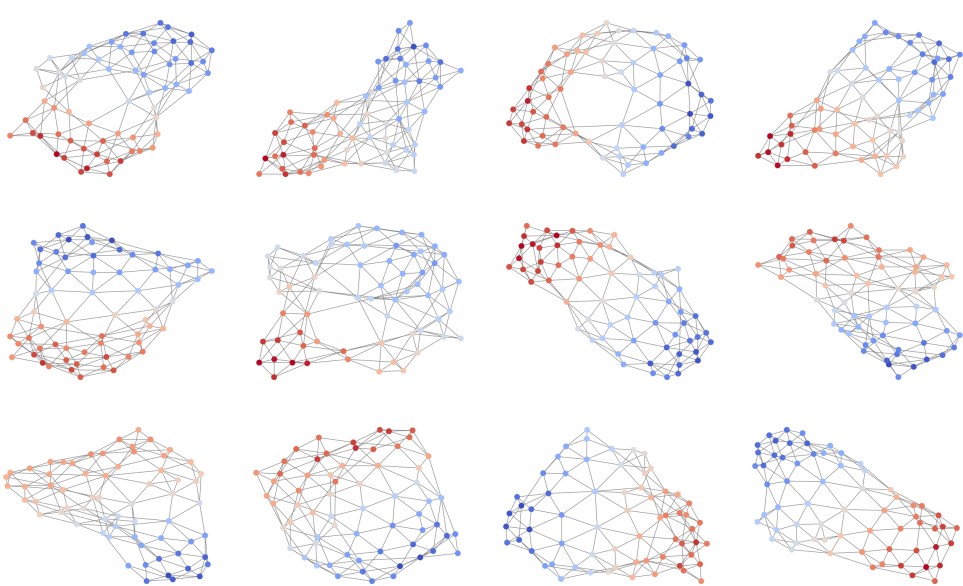

Figure 6: The generated graphs from the Planar dataset.

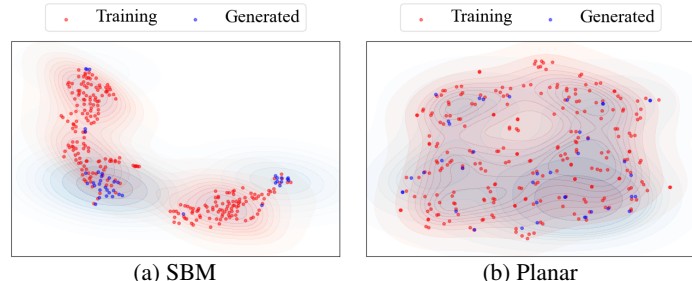

(a) SBM                      (b) Planar

Figure 7: The t-SNE visualization of the Planar and SBM datasets, along with generated representations by our framework. Each point denotes the representation of a training graph sample or from our representation generator.

solution is to create a dataset that is directly generated from representations. In this manner, the representations learned in our framework could be evaluated for their quality and similarity to the ground truths.

# H BROADER IMPACT

As our datasets only involve datasets, the information in these datasets is ensured to be anonymized. We only use datasets from public releases, and thus we infer that this work does not have negative social impacts.

