# OpenReview forum: "GraphRCG: Self-Conditioned Graph Generation"
_ICLR.cc/2025/Conference — Submitted to ICLR 2025_

### Official Review · Reviewer_vjaU · 2024-10-29

**Soundness:** 2
**Presentation:** 2
**Contribution:** 2
**Rating:** 3
**Confidence:** 4

**Summary:**

The work presents a two-step approach to graph generation. First, a vector representation is generated, which conditions the subsequent graph generation.
These vector representations are obtained using a graph encoder, jointly trained with a generative diffusion model that transforms noise into the representations.
Graph generation is performed by a conditional diffusion model, parameterized by a message-passing transformer, that generates graphs conditioned on representations.
The model is evaluated on several graph generation benchmarks and shows competitive performance compared to existing methods.

**Strengths:**

The paper is well structured and easy to follow. Figure 2 provides a clear overview of the proposed method and its components. The work introduces some interesting ideas, such as the joint training of the graph encoder and the generative diffusion model, and alignment losses during both representation and graph generation training. The empirical performance of the proposed method is demonstrated on several graph generation benchmarks, showing competitive performance compared to existing methods. Additionally, the ablation study offers insights into the significance of the various components of the method.

**Weaknesses:**

First of all, there appears to be a problem with the latex rendering in the paper, as a substantial portion of symbols in math equations are missing. So the following comments are based on what could be inferred from the text and figures.

1) **Contributions and differences from previous work**:
The paper adapts the previous work RCG [1] from image to graph generation. However, there are key differences: while RCG employs a pretrained image encoder to obtain representations, the proposed method jointly trains the graph encoder with a diffusion model. Additionally, the proposed method introduces alignment losses during both representation and graph generation training. Furthermore, during graph generation, the diffusion model is conditioned on a noisy version of the representation, rather than the "clean" one, with noise strength aligning with the current denoising step. The paper should clearly delineate which aspects of the method are adapted from [1] and which are novel.

2) **Theoretical justification**:
The joint training, alignment losses, and noisy conditioning convolute the modeling approach. For instance, if the graph generation diffusion model were conditioned on the clean representation, it would be clear to see that the model distribution p(G) is decomposed as P(G) = P(G | R) P(R), where P(R) means the distribution of the representation R. However, with the noisy conditioning, it is not clear what the model distribution decomposes into, and what the role of the conditioning noise is. The alignment losses add further complexity. Although the ablation study demonstrates empirical benefits of these components, there is neither theoretical justification nor intuitive explanation for these design choices, making it difficult to grasp the underlying principles of the proposed method. Similarly, the joint training of the graph encoder and the representation diffusion model lacks theoretical justification or intuitive explanation for why it would yield meaningful representations. Additionally, as the representation distribution evolves during generator training, the convergence of this procedure should be discussed.

3) **Fair ablation study**:
The ablation study highlights the empirical benefits of the method's components, but the increase in model capacity and training time is ignored. For instance, the total number of parameters for the generative model is X (parameters for the representation diffusion model) + Y (parameters for the graph generation diffusion model). It would be insightful to see how the performance of the proposed method compares to an unconditional graph generation diffusion model with the same number of parameters, i.e. X + Y. Furthermore, for a fair comparison, the training time of the proposed method (joint training of the graph encoder and the representation diffusion model + conditional graph generation training) should be compared to other graph generation methods.

4) **Related work**:
Some overlooked works on graph generation should be discussed in the related work section: [2-8].


[1]: Li et al., Return of Unconditional Generation: A Self-supervised Representation Generation Method

[2]: Goyal et al., GraphGen: A Scalable Approach to Domain-agnostic Labeled Graph Generation

[3]: Grover et al., Graphite: Iterative Generative Modeling of Graphs

[4]: Davies et al., Size Matters: Large Graph Generation with HiGGs

[5]: Haefeli et al., Diffusion Models for Graphs Benefit From Discrete State Spaces

[6]: Diamant et al., Improving Graph Generation by Restricting Graph Bandwidth

[7]: Qi et al., SwinGNN: Rethinking Permutation Invariance in Diffusion Models for Graph Generation

[8]: Bergmeister et al., Efficient and Scalable Graph Generation through Iterative Local Expansion

**Questions:**

1) Figure 1:
From this figure, one could infer that the representation and graph generation diffusion models are trained separately and only during graph generation is the conditional information incorporated. This interpretation contradicts the description in the rest of the paper.
Additionally, it is unclear what the term *capture* in this context means and how it differs from "learning".
I suggest revising the figure to better reflect the overall approach.


2) Validity of planar and sbm graphs:
Previous studies (SPECTRE and DiGress) report the fraction of valid planar and sbm graphs generated. Given that Table 2 includes results from these studies, why has the validity of planar and sbm graphs been omitted? How does the proposed method compare in this regard?

---

### Official Review · Reviewer_6DVd · 2024-10-31

**Soundness:** 3
**Presentation:** 2
**Contribution:** 2
**Rating:** 3
**Confidence:** 5

**Summary:**

The paper proposes GraphRCG, a self-conditioned framework for graph generation that leverages graph distributions explicitly to enhance graph generation quality and fidelity. Unlike conventional models that implicitly capture graph distributions through direct output alignment, GraphRCG encodes each graph into a low-dimensional representation. These representations are then used to guide the generation process progressively. Extensive experiments demonstrate GraphRCG's superior performance compared to baseline models​.

**Strengths:**

- The model corporate the implicit representation distribution into modeling graph data is reasonable.
- Experiments show the superior performance compared to other baselines.
- This paper is easy to follow and the motivation is clear.

**Weaknesses:**

- Although the motivation is clear, the improvement compared to other conditional graph generative models seems is incremental. Augmenting the data feature and modeling joint distribution of $p(G, z)$ with representation in latent space are common in generative models, not limited to graph generation and the learnable features, so is there any comparison between modeling the joint distribution and inject the additional feature into the original model.

- I strongly suggest you should compare proposed model with more advanced graph generative models, like EDGE[1], SwinGNN[2], GruM[3] and so on.
- Have you tried other more complex neural network architecture like graph transformer in DiGress[4]?



[1] Efficient and degree-guided graph generation via discrete diffusion modeling.

[2] SwinGNN: Rethinking Permutation Invariance in Diffusion Models for Graph Generation.

[3] Graph generation with diffusion mixture.

[4] Discrete denoising diffusion for graph generation.

**Questions:**

- Is there something wrong in the formula rendering in the submitted pdf? It cause great inconvenience to read this paper.

---

### Official Review · Reviewer_naAA · 2024-11-02

**Soundness:** 1
**Presentation:** 1
**Contribution:** 2
**Rating:** 3
**Confidence:** 4

**Summary:**

This paper introduces GraphRCG, a self-conditioned graph generation framework to align the generated graph distribution to the training distribution. The authors propose to encode graphs into low-dimensional representations so that the graph encoders capture the data distribution and then leverage the learned representations to guide the generation process. Additionally, GraphRCG introduces step-wise guidance where the generative process of the representations progressively affects the graph generative process. The authors perform experiments on generic and molecular graph datasets.

**Strengths:**

- The proposed method improves performance by leveraging the auxiliary graph representations in the generative process.
- Compared to the original RCG [1], the proposed step-wise guidance is novel.

[1] Li et al., "Return of Unconditional Generation: A Self-supervised Representation Generation Method", NeurIPS 2024.

**Weaknesses:**

- The proposed techniques are largely covered by recent works. The graph generation technique is directly adopted from DiGress [2] which also utilizes the marginal transitions and conditioning method with graph features. Additionally, conditioning the generative process with the generated representations is proposed in the original RCG [1].

- The second challenge lacks soundness and clarity. The authors argue that graph generation is inherently sequential, as meeting the desired conditions should be performed in a sequential manner. However, to meet the desired conditions, accurately generating core substructures, such as communities in general graphs or functional groups in molecules, is crucial and it is difficult to be achieved by using a sequential approach. Therefore, directly guiding the generation toward the true distribution seems to be promising to meet the desired properties. I suggest that the authors clarify why the step-wise guidance is crucial for graph generation.

- The experimental setup does not show that the learned graph representations can actually capture the data distributions. To show this effect, the authors should show whether the learned representations can understand the structural information in the training dataset. For example, on the SBM dataset, it would be better to classify the training and generated graphs in terms of the number of communities and then visualize the corresponding representations of training and generated graphs.

- The experimental result in section 4.2 does not show the interpolation effect, as the target graph distribution is not diverse. I recommend interpolating between distinct graphs such as molecules with benzene rings and molecules without benzene rings.

- There are significant errors in most of the equations, making it difficult for readers to understand the proposed method.

- Some related works such as EDGE [3] and GruM [4] are missing and not compared. Also, it would be better to apply the proposed method to the recent diffusion models to emphasize the effectiveness of the proposed method.

[1] Li et al., "Return of Unconditional Generation: A Self-supervised Representation Generation Method", NeurIPS 2024.

[2] Vignac et al., "DiGress: Discrete Denoising diffusion for graph generation", ICLR 2023.

[3] Chen et al., "Efficient and Degree-Guided Graph Generation via Discrete Diffusion Modeling", ICML 2023.

[4] Jo et al., "Graph Generation with Diffusion Mixture", ICML 2024.

**Questions:**

- Is the discrete diffusion model actually more beneficial than the diffusion model on the continuous space such as GDSS or GruM for the proposed framework?

- Which architectures did the authors use for the graph encoder?

- Why did the authors not use the pre-trained graph encoders?

---

### Official Review · Reviewer_8H7x · 2024-11-03

**Soundness:** 3
**Presentation:** 2
**Contribution:** 2
**Rating:** 5
**Confidence:** 4

**Summary:**

This paper proposes a self-conditioned graph generation method for capturing the entire distribution. The representation generator produces a low-dimensional representation of graphs that reflect the learned distribution, and the learned distribution is used to guide the denoising diffusion process. The experimental results show that representation-generated conditioning improves generation quality.

**Strengths:**

- The motivation of this work seems reasonable: Using graph representation for conditioning the generation process to learn the complex distribution. Yet, this motivation and the method of using representation generation has been studied in previous work (Li et al., 2024)

Li et al., 2024, Return of Unconditional Generation: A Self-supervised Representation Generation Method

**Weaknesses:**

- This paper is an unfinished work as the equations and notations are incomplete. For example, as in Eq.(5), (6), (10), and (11), random variables are only represented as subscripts and the parenthesis are in weird positions.

- The idea of using a representation generator to condition the generation process is mostly based on the Representation-Conditioned Generation framework (Li et al., 2024) which conditions the image generator with generated representation from the self-supervised encoder. The difference only comes from the difference in the domain.

- The V.U.N. (valid, unique, and novel) metric for generic graph generation (SBM, Planar) is required to fully evaluate the model.

- The experimental results are different from those reported in previous works: For SBM and Planar, the DiGress results in this paper are worse than the original results from the DiGress paper. For the molecule generation task, the results of DiGress are worse than the results reported in the recent work (Jo et al., 2024); DiGress and GruM (Jo et al., 2024) achieve very low FCD in QM9 and ZINC250k (DiGress: QM9=0.095, ZINC250k=3.482, GruM QM9=0.108, ZINC250k=2.257). What is the reason for this difference?

[1] Li et al., Return of Unconditional Generation: A Self-supervised Representation Generation Method, arXiv 2024

[2] Jo et al., Graph Generation with Diffusion Mixture, ICML 2024

**Questions:**

Please address the weaknesses above.

---

### Meta-Review · Area_Chair_K9th · 2024-12-16

**Metareview:**

All reviewers agree that the method of this submission is mainly based on recent work and there are many holes in writing and experiments. The authors did not provide any feedback in the rebuttal and discussion phase. Taking all these into account, I think this submission is not qualified enough for ICLR.

**Additional Comments On Reviewer Discussion:**

All the reviewers provide their comments. Their concerns are consistent, including the novelty of the method, the writing quality, and the solidness of experiments. After reading their comments and the paper, I agree with their judgments.

---

### Decision · Program_Chairs · 2025-01-22

Reject